# ESBL *Escherichia coli* Isolates Have Enhanced Gut Colonization Capacity Compared to Non-ESBL Strains in Neonatal Mice

Aspen Kremer,[a] Grant Whitmer,[b] Alondra Diaz,[a] Alima Sajwani,[a] Alexis Navarro,[c] Mehreen Arshad[a,b]

[a]Ann and Robert H. Lurie Children's Hospital, Chicago, Illinois, USA
[b]Feinberg School of Medicine, Northwestern University, Chicago, Illinois, USA
[c]University of North Carolina, Chapel Hill, North Carolina, USA

Aspen Kremer and Grant Whitmer contributed equally to this article. Authors are in order of the effort contributed towards the data collection and manuscript preparation.

**ABSTRACT** Extended-spectrum beta-lactamase (ESBL)-producing *Escherichia coli* can cause invasive infections in infants and immunocompromised children with high associated morbidity and mortality. The gut is a major reservoir of these strains in the community. Current dogma dictates that antimicrobial resistance is associated with a fitness cost. However, recent data show that some contemporary ESBL *E. coli* strains may be more "fit" compared to nonresistant *E. coli* strains. Here, we use whole-genome sequencing to first characterize 15 ESBL *E. coli* strains isolated from infants in a Pakistani community, a clinical extraintestinal pathogenic ESBL *E. coli* ST131 strain, and a non-ESBL commensal *E. coli* strain, and then use a novel animal model of early life gut colonization to assess the ability of these strains to colonize the infant mouse gut. We determined that CTX-M-15 was present in all the ESBL strains, as well as additional beta-lactamases and genes conferring resistance to multiple antibiotic classes. In the animal model, 11/16 ESBL *E. coli* strains had significantly higher burden of colonization at week four of life compared to commensal strains, even in the absence of selective antibiotic pressure, suggesting that these strains may have enhanced fitness despite being highly antimicrobial resistant.

**IMPORTANCE** Antimicrobial resistance is a global public health emergency. Infants, especially preterm infants and those in the neonatal intensive care unit, immunocompromised hosts, and those with chronic illnesses are at highest risk of adverse outcomes from invasive infections with antimicrobial-resistant strains. It has long been thought that resistance is associated with a fitness cost, i.e., antimicrobial-resistant strains are not able to colonize the gut as well as nonresistant strains, and that antibiotic exposure is a key risk factor for persistent colonization with resistant strains. Here, we use a novel infant mouse model to add to the growing body of literature that some highly-resistant contemporary *Escherichia coli* strains can persist in the gut with a significant burden of colonization despite absence of antibiotic exposure.

**KEYWORDS** enterobacteriaceae, *Escherichia coli*, antibiotic resistance, intestinal colonization, neonates

Address correspondence to Mehreen Arshad, marshad@luriechildrens.org.

The authors declare no conflict of interest.

Antibiotic resistance among Gram-negative bacteria (GNB) are an evolving threat with clinical consequences (1). GNB such as Enterobacterales, especially *Escherichia coli*, are also a leading cause of neonatal infections, among preterm infants as well as term infants in the neonatal intensive care unit (2–4). In developing countries, mortality is higher in neonates and infants infected with extended-spectrum beta-lactamase (ESBL)-producing Enterobacterales, likely attributed to a delay in appropriate therapy (5). In developed countries, infections with these strains significantly increase

neonatal morbidity, length of hospital stay, and health care costs (6). The disability-adjusted life-years (DALYs) for infants <1 year infected with ESBL-producing Enterobacterales are significantly higher than for any other age group (7).

There are multiple molecular mechanisms of antibiotic resistance among GNB (8, 9), of which widespread production of ESBLs is of clinical concern. ESBL-producing Enterobacterales, such as *Escherichia coli* strains, are identified worldwide and can colonize healthy individuals without antibiotic exposure (10). While some ESBL *E. coli* appear to have an associated fitness cost (11, 12), others carry genes that may convey competitive advantage (13, 14). One example of such genes is fimbriae, which have been implicated to be essential for adherence and invasion of gut colonization (15). Current studies on gut colonization largely focus on *E. coli* strain ST131 or are cross-sectional analyses of human cohorts. Little is known about persistent gut colonization with non-ST131 ESBL *E. coli* strains.

In our recent study (10), we isolated ESBL *E. coli* strains of distinct sequence types from the gut of otherwise healthy Pakistani infants. We now report data from whole-genome sequencing (WGS) on a subset of these strains, describe their metabolic gene profile, and use a mouse model to show that some ESBL strains result in increased burden of colonization compared to commensal strains. We postulate this increased burden may be due to differences in copy numbers of key metabolic genes and the presence of certain colonization factors.

## RESULTS

**Phylogenetic analysis.** In total, we conducted WGS of 18 isolates. Among these, there were 16 ESBL isolates obtained from healthy Pakistani infants, which showed 13 distinct sequence types. We also reported three new sequence types, ST8130, ST8131, and ST8136, that had not been previously reported. In addition, we sequenced a non-ESBL isolate (ST6438) that was also obtained from a healthy infant and used that as a control strain, and an ST131 strain obtained from the clinical microbiology lab at Lurie Children's Hospital. ST131 was not isolated from any of the infants but was included in this study given its global prevalence. Fig. 1 shows a dendrogram of the sequence type and phylogroup of the strains. The accession number of all the strains can be found in Table S1 in the supplemental material.

**Resistance and virulence genes among ESBL *E. coli* isolates.** Multiple beta-lactamase genes were present in the ESBL *E. coli* isolates, with CTX-M-15 present in all ESBL *E. coli* strains (Fig. 1). All ESBL *E. coli* strains also carried genes conferring resistance to multiple other classes of antibiotics.

All ESBL *E. coli* strains were positive for *csg* (curli fiber-encoding gene). All strains except ST1201, ST8130, ST8131, and ST8136 were positive for *fim* genes. Other virulence genes associated with gut colonization included *ecp* or *yag* (encoding *E. coli* common pilus), and *aap* (dispersin aggregated protein). The *aap* gene was only found in ST8131 and ST38.

**Metabolic gene profile.** A heat map displaying the copy numbers of genes related to glycolysis, the tricarboxylic acid cycle (TCA), and oxidative phosphorylation across strains is shown in Fig. 2. Both ST28 and ST410 had significantly higher copy numbers of several metabolic genes. In addition, compared to the two comparator strains, the ESBL strains adept in gut colonization also had on average higher copy numbers of several key metabolic enzymes associated with the carbohydrate metabolism (fructose-bisphosphatase, 6-phospho-beta-glucosidase, and fructose-bisphosphate aldolase), the TCA cycle (fumarate hydratase), and aerobic and anaerobic respiratory chain (NADH:ubiquinone reductase).

**Bacterial strain growth.** All strains used in these experiments showed a similar growth pattern *in vivo* when grown in LB. This was noted by using optical density measures and confirmed by growth on LB agar with selective antibiotics.

Similarly, to ensure the stability of the ESBL gene, all ESBL *E. coli* strains were also grown in liquid LB for up to 3 weeks (21 days), with daily subcultures in the absence of any antibiotics and CFU/mL of ESBL *E. coli* for each strain noted on day 1 and day 21

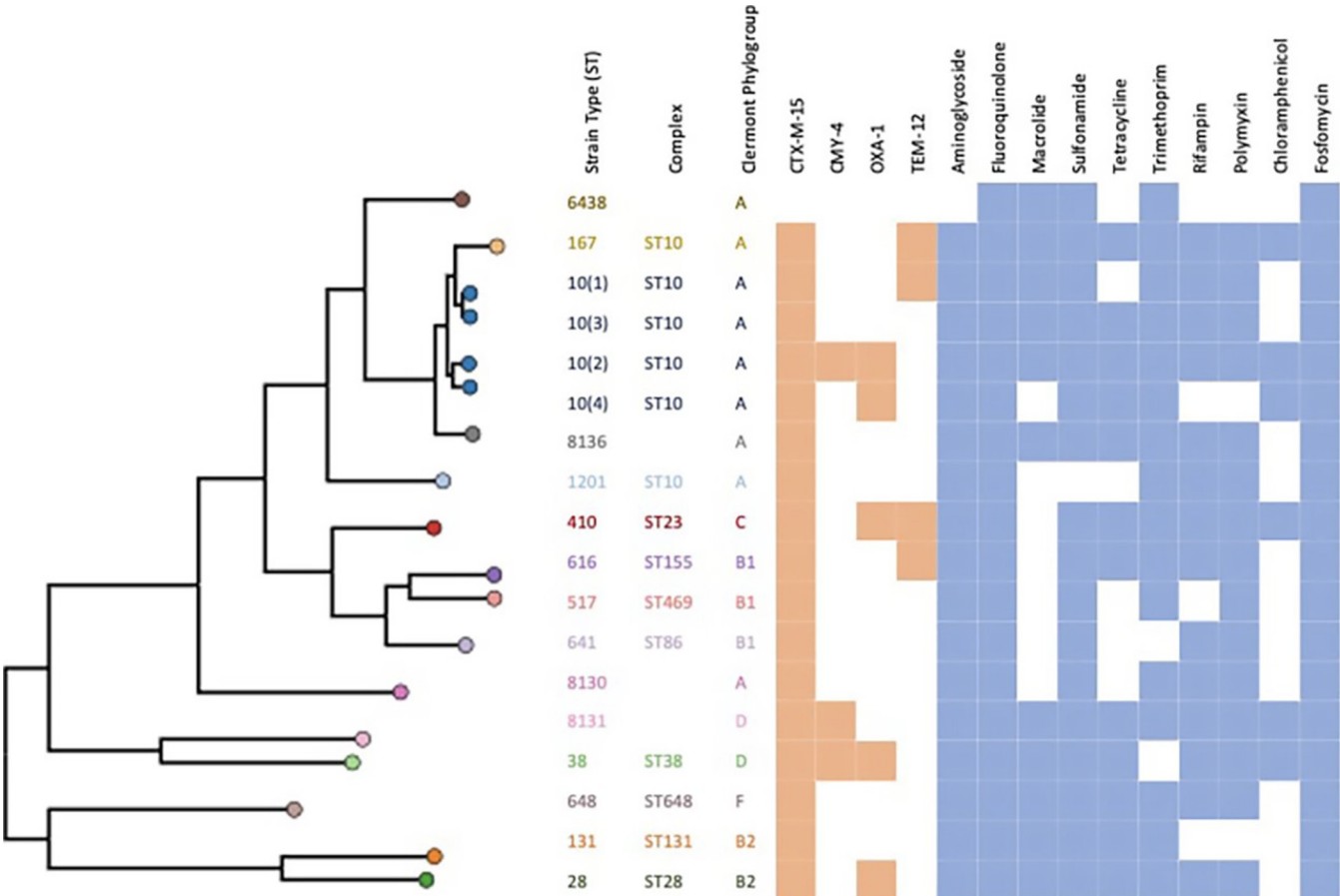

**FIG 1** Dendogram of all the strains included in this study along with their phylogroup, known virulence factors associated with gut epithelium adhesion, beta-lactamase genes found in each strain, and the genes conferring resistance against multiple classes of antibiotic. Colored boxes represent presence of the relevant genes (green, virulence factors; orange, beta-lactamase genes; blue, multiple classes of antibiotics).

by selecting on LB agar with ceftriaxone. We noted no difference in the CFU/mL between these two time points.

**Murine model.** At 3 weeks of age, strains ST10(3), ST410, ST648, ST8131, and ST8136 had significantly higher bacterial CFU/g counts compared to non-ESBL-producing strain ST6348 (Fig. 3A). After these pups were weaned from mouse milk to regular chow *ad libitum*, several other strains were also found to have significantly higher burden of colonization than the commensal strains. At week four, these included strains ST10(2), ST10(4), ST28, ST38, ST167, ST410, ST517, ST616, ST641, ST8131, and ST8136 (Fig. 3B). We noted that this difference was largely due to a significant decrease in colonization burden with both of the non-ESBL *E. coli* strains at week 3 and week 4: MG1655 (average CFU/g $1.2 \times 10^5$ versus $5.9 \times 10^3$, $P = 0.02$) and ST6438 (average CFU/g $6.1 \times 10^5$ versus $7.3 \times 10^3$, $P = 0.0001$). At the same time, there was no significant difference in the colonization burden of most ESBL *E. coli* strains between the two time points. There were no gross abnormalities in the animals at any time point, and we noted no significant difference in weights between the animals inoculated with the different strains. Control pups ($n = 6$) as well as dams ($n = 2$) that had no intervention did not have any *E. coli* in the feces.

**Competition experiments.** When pups were given a mixture with equal amounts of an ESBL *E. coli* strain and the non-ESBL-producing *E. coli* strain ST6438, the ESBL strains ST28 and ST410 had significantly higher bacterial CFU/g counts compared to ST6438 at three and four of weeks age (Fig. 4). In contrast, ST6438 had significantly higher bacterial CFU/g counts compared to ESBL-producing ST1201 at three and four weeks of age, as was noted in the single isolate inoculation experiments shown in Fig. 3.

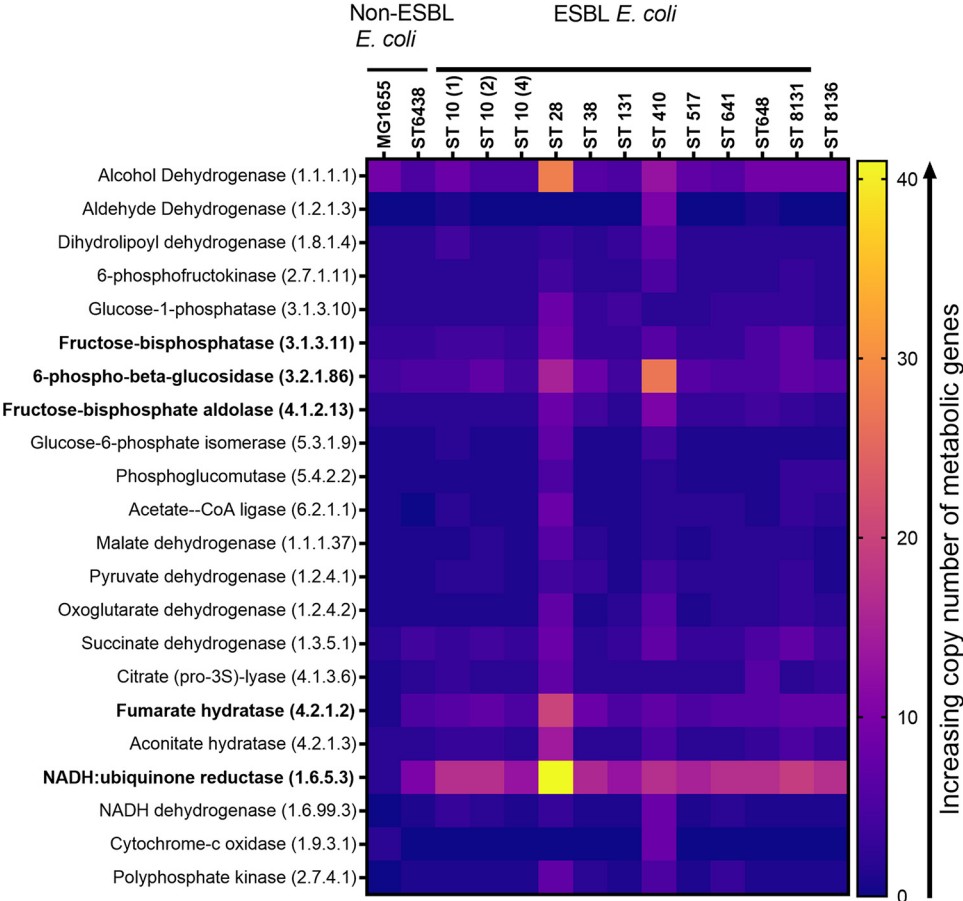

**FIG 2** Heatmap showing the copy numbers of key metabolic genes in *E. coli*. Several genes (in bold) had higher copy number in ESBL *E. coli* compared to non-ESBL *E. coli*. ST28 and ST410 carried significantly more metabolic genes than all other strains, especially the non-ESBL strains.

## DISCUSSION

ESBL *E. coli* strains can be passed on to infants by perinatal transmission, colonize the infant gut, and subsequently increase the risk of invasive infections in infants in the early life period (16–19). At the same time, asymptomatic infants also contribute significantly to the community reservoir of these strains. Healthy infants in some communities have ESBL-producing Enterobacterales colonization rates ranging from 43 to 63% (10, 20, 21). However, most of these human-cohort studies are cross-sectional studies, and persistent gut colonization with ESBL *E. coli* after acquisition in the early life period, and in the absence of antibiotic or environmental exposure, has not been studied. We therefore developed an animal model to systematically investigate this phenomenon. We were especially interested in investigating ESBL *E. coli* from asymptomatic infants because these strains would likely have the potential to silently persist for prolonged periods of time, contributing to the reservoir of the ESBL genes within the community. Investigating such strains in the relevant mouse model may, in the future, allow development of meaningful strategies to reduce the overall burden of the ESBL *E. coli* in the community.

Fourteen of the sixteen ESBL *E. coli* isolates obtained from healthy Pakistani children belonged to unique sequence types from four common phylogenetic groups (A, B1, B2, and D), and one each belonged to group C and group F, suggesting considerable diversity of ESBL *E. coli* colonization in the community infants, instead of a predominance of specific sequence types. Most commensal and diarrheagenic *E. coli* strains belong to Group A, most extraintestinal pathogenic (ExPEC) strains belong to Group

**A**

Week 3

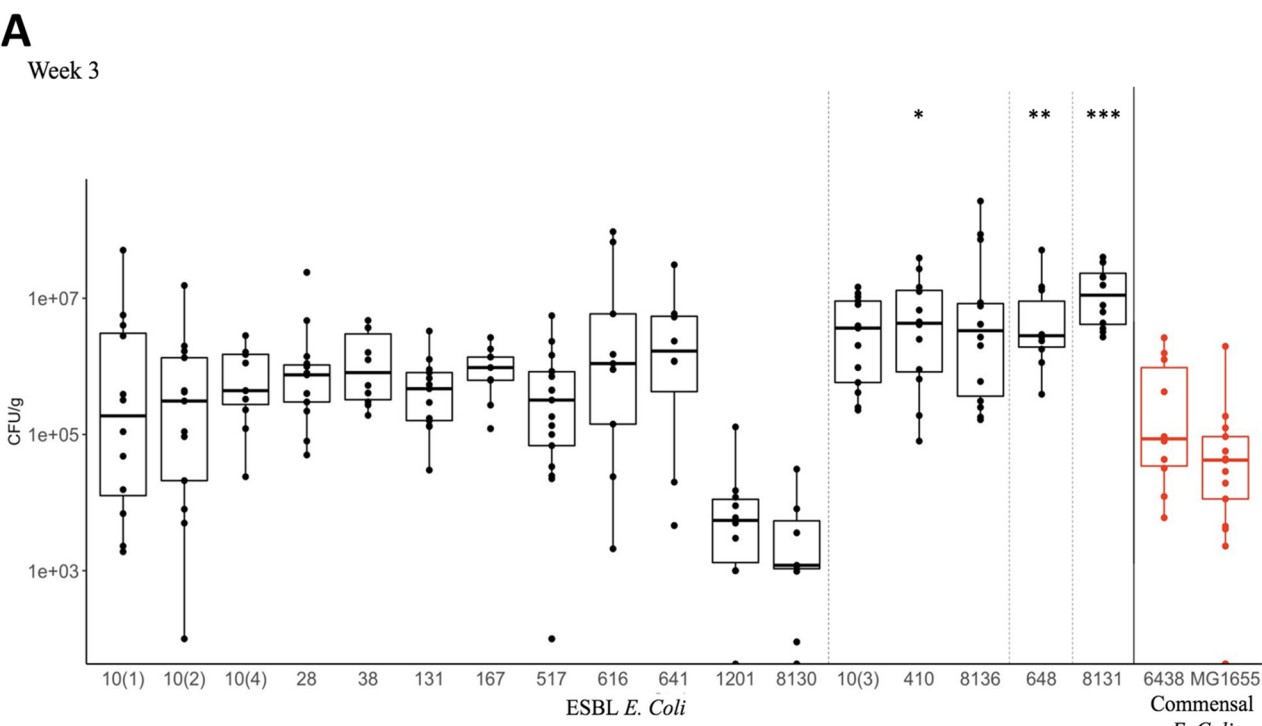

Alpha was set at 0.025; * = p ≤ 0.025, ** = p ≤ 0.001 ,*** = p ≤ 0.0001

**B**

Week 4

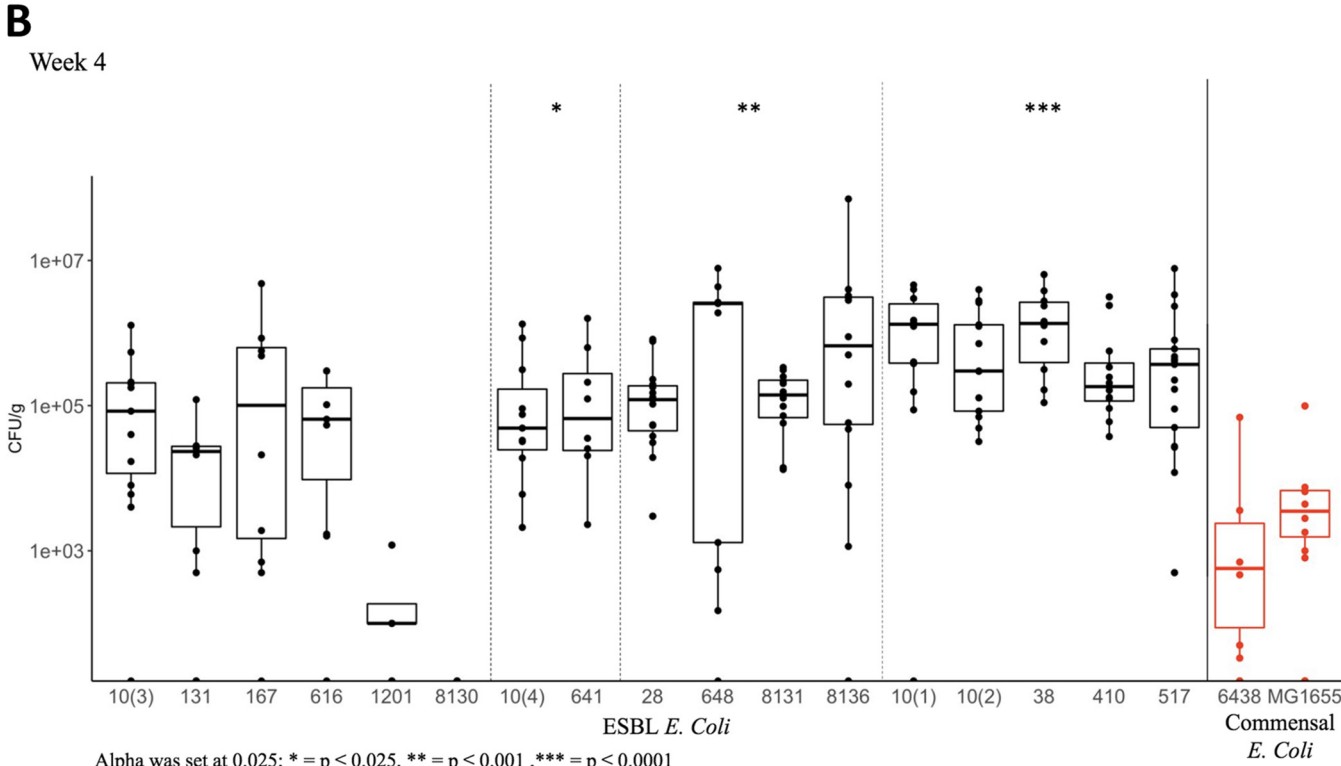

Alpha was set at 0.025; * = p ≤ 0.025, ** = p ≤ 0.001 ,*** = p ≤ 0.0001

**FIG 3** The results of the gut colonization experiments at (A) 3 weeks and (B) 4 weeks of life, respectively. Each strain is denoted with the same color in both of the graphs. Several ESBL *E. coli* strains had significantly higher burden of colonization than ST6438 (non-ESBL *E. coli*) and MG1655 (lab-commensal *E. coli*). Statistical test: Dunn's test, with alpha set at 0.025; *, $P \leq 0.025$; **, $P \leq 0.001$; ***, $P \leq 0.0001$.

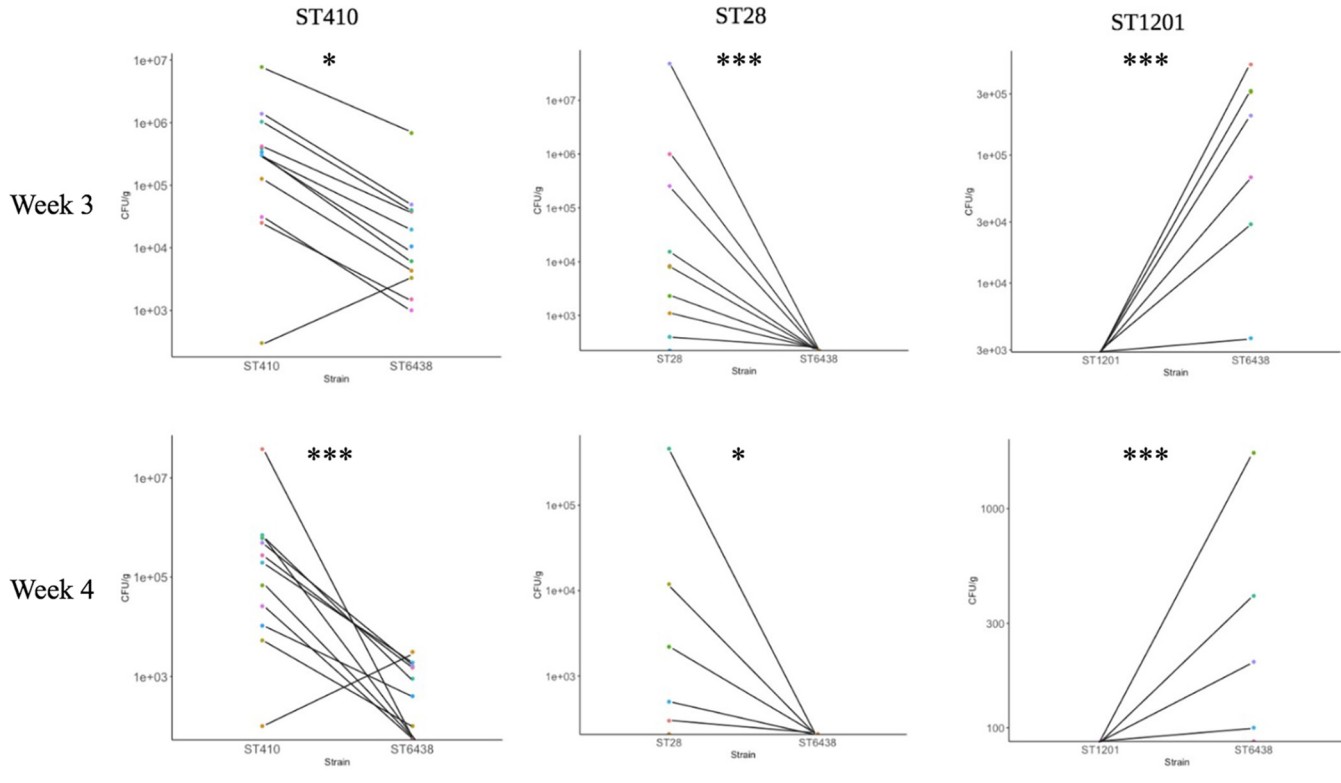

Alpha was set at 0.025; * = p ≤ 0.025, ** = p ≤ 0.001 ,*** = p ≤ 0.0001

**FIG 4** The results of the *in vivo* competition experiments between the commensal strain ST6438 and the ESBL *E. coli* strains ST410, ST28, and ST1201, respectively, at weeks 3 and 4. Each circle represents a mouse, and the straight line denotes the difference in bacterial CFU/g between the respective ESBL *E. coli* strains and the commensal strain in the same mouse at a given time point. Statistical test: Mann-Whitney-Wilcoxon test was used; *, $P \leq 0.025$; **, $P \leq 0.001$; ***, $P \leq 0.0001$.

B2, and Group D mostly consists of commensals and some ExPEC strains (22). Some clinically important sequence types were noted among the isolates. ST10 is known to be ubiquitous in human fecal and food samples and is being increasingly linked to high-level resistance globally (23). ST28 is associated with invasive disease in the pediatric age group (24), and ST410 is a globally prevalent ExPEC strain that is considered a "high-risk" clone due to associated antimicrobial resistance genes and its successful clonal expansion (25). There were three sequence types that had not been previously reported: ST8130, ST8131, and ST8136.

All strains had CTX-M-15, the most common ESBL gene worldwide (26, 27). Other common ESBLs included TEM-12 and OXA-1; both can be cotranscribed with the CTX-M beta-lactamases. These strains also carried multiple genes conferring resistance against other classes of clinically important antibiotics, indicating that the infant gut is a prominent reservoir of antibiotic-resistant *E. coli* strains. Importantly, we noted that the strains did not lose resistance to ceftriaxone over several weeks. This is perhaps not surprising since the CTX-M-15 ESBL gene is often associated with incompatibility type-F (IncF) plasmids (28, 29). These plasmids are known to have several mechanisms, such as toxin–antitoxin addiction systems, due to which they remain stable within the host organism even in the absence of selective antibiotic pressure (30). Whole-genome sequencing of the pathogenic *E. coli* strains yielded information that may explain the enhanced colonization ability *in vivo*. Genes known to be involved in curli production were found in all isolates. All strains except ST1201 and ST8130, the poorest colonizers, had at least one other virulence factor (*fim* or *ecp* genes) associated with gut colonization (31, 32), suggesting that in the absence of these adherence genes, ESBL *E. coli* are poor gut colonizers.

Using the Comparative Pathway tool in PATRIC, we interrogated the metabolic gene

profile of the ESBL *E. coli* strains that had colonized the mouse gut with a significantly higher burden of colonization compared to the non-ESBL strains MG1655 and ST6438. This tool compares pathways across genomes using KEGG pathway maps. Several crucial genes tended to have higher copy numbers in the ESBL strains, such as 6-phospho-beta-glucosidase (hydrolyzes a variety of glucosides [33]), fructose bisphosphatase (important for regulation of gluconeogenesis [34]), fructose-biphosphate aldolase (central regulator in glycolysis and gluconeogenesis pathways [35]), fumarate hydratase (key contributor to *E. coli* fitness under iron limitation [36]), and NADH:ubiquinone reductase (central regulator of both the aerobic and anaerobic respiratory chain [41]). Two ESBL *E. coli* strains (ST28 and ST410) also had significantly higher copy number of metabolic genes than all other strains, including non-ESBL strains. These metabolic genes should be investigated as contributors to the competitiveness of ESBL-producing strains over their non-ESBL counterparts, which has been previously reported (14, 37). The metabolic profile of bacterial strains may be especially relevant in the immediate postpartum period, when limited nutrients are available in the neonatal gut. ESBL Enterobacterales, with a higher metabolic capacity and enhanced ability to utilize scarce nutrients, may thus be able to outcompete non-ESBL strains.

In our mouse model, most ESBL *E. coli* remained unaffected by the change in mouse diet from mouse milk at the time of weaning (3 weeks of life) to regular chow, which suggests that these strains would remain persistent colonizers even as an infant diet evolves over the early life period. This is unlike the non-ESBL *E. coli* strains, where there was a significant reduction in the colonization burden 1 week after weaning from the mouse milk, a time when the gut microbiome is thought to increase in diversity in both humans (38) and mice (39).

The *in vivo* competition experiments further supported our observation that some ESBL *E. coli* are better gut colonizers that non-ESBL *E. coli* strains. Those that had a higher burden of colonization when inoculated individually (ST28 and ST410) were able to out-compete the commensal ST6438 strain, whereas the ESBL *E. coli* strain ST1201, which was a poor colonizer in the initial experiments, was outcompeted by ST6438 in the competition experiment.

In summary, our work indicates that ESBL *E. coli* strains isolated from otherwise healthy infants often carry genes conferring resistance against numerous classes of antibiotics, have virulence genes known to play a role in gut colonization, and have higher copy numbers of important metabolic genes. Our mouse model of early life intestinal colonization demonstrated increased *in vivo* colonization ability in some ESBL strains.

Limitations of this study include potential differences between *E. coli* colonization in mice and humans, and the isolation of our strains from a single geographic area.

## MATERIALS AND METHODS

**Sample collection, selective markers, and bacterial growth.** We have previously conducted a cross-sectional study to determine the burden of gut colonization with ESBL Enterobacterales among otherwise healthy children residing in four peri-urban communities in Karachi, Pakistan (10). Stool samples were collected from 5- to 7-month-old infants in the community that had no known exposure to antibiotics and did not have any illnesses in the month prior to stool sample collection. Unfortunately, maternal data were not reliably available. Stool samples were plated on MacConkey agar at a microbiology lab at Aga Khan University, Karachi, Pakistan, and the phenotypic resistance pattern was determined, as has been reported before (10). Individual ESBL and some non-ESBL *E. coli* as controls were isolated and frozen at −80°C at the local Pakistani lab. They were then transported to Lurie Children's Hospital in Chicago, USA on dry ice for further analysis.

The strain ST131 was obtained separately from the clinical microbiology lab at Lurie Children's Hospital. The initial sequencing typing was done using the Pasteur method, and then confirmed using whole-genome sequencing as detailed below (40). A non-ESBL-producing commensal *E. coli* strain was also isolated from a healthy infant's stool from the same infant cohort mentioned above; however, this infant was not a carrier of any ESBL Enterobacterales. A spectinomycin selective marker was added using P1 transduction following a previously published protocol (41). MG1655, a lab commensal strain, was similarly marked with the kanamycin resistance marker. Intrinsic ceftriaxone resistance was used for the ESBL-producing *E. coli* strains.

Unless stated otherwise, we grew bacterial cultures at 37°C and under agitation (225 rpm) in lysogenic broth (LB) medium overnight, supplemented with appropriate amounts of antibiotics (20 $\mu$g/mL ceftriaxone, 50 $\mu$g/mL spectinomycin, 50 $\mu$g/mL kanamycin). We stored isolates in 25% glycerol at −80°C.

All strains showed a similar growth pattern *in vitro* when grown in LB. This was noted by using optical density measures and confirmed by growth on LB agar with selective antibiotics. Similarly, the ESBL *E. coli* strains were also grown in liquid LB for up to 3 weeks (21 days), with daily subcultures in the absence of any antibiotics, and CFU/mL of ESBL *E. coli* for each strain was noted on day 1 and day 21. We noted no difference between these two time points.

**Whole-genome sequencing and analysis.** DNA was extracted using the Wizard Genomic DNA purification kit (Promega, Madison, WI, USA) and submitted for whole-genome sequencing at the Northwestern Sequencing core facility. Briefly, DNA was quantified using Qubit (Thermo Fisher Scientific, Waltham, MA, USA). DNA-seq libraries were made using Kapa Hyper Prep kit (Roche, Pleasanton, CA, USA). Libraries were sequenced using HiSeq 4000 at 300-bp paired ends. The fastq sequences were assembled *de novo* using standard pipelines in EnteroBase (http://enterobase .warwick.ac.uk/species/index/ecoli) (42). Achtman multilocus sequence typing (MLST) was conducted using these annotated genomes (using prokka v1.11) and the MLST pipeline (v3.2) in EnteroBase. To understand the genetic relatedness, we constructed a single-nucleotide polymorphism (SNP) tree using the EnteroBase pipeline. All assembled genomes were submitted to the National Center for Biotechnology Information (NCBI).

**Identification of resistance, virulence, and metabolic genes.** The Pathosystems Resource Integration Center (PATRIC) (43) was used to perform a BLAST-based comparison using PATRIC's collection of antibiotic resistance proteins and the externally curated Comprehensive Antibiotic Resistance Database (CARD) (44). Similarly, the Comprehensive Genome Analysis tool within PATRIC was used to identify virulence factors that are known to play a role in adhesion to gut epithelium and therefore play a role in gut colonization (curli fibers, fimbriae, pilus, and dispersin aggregated proteins [45]). This tool also uses BLAST-based comparisons of annotated genome with several databases, including the PATRIC virulence factor database (PATRIC_VF) and the externally curated databases Victors (a web-based resource for virulence factors in human and animal pathogens) and Virulence Factor DataBase (VFDB). For all BLAST-based comparisons, the default settings within PATRIC were used, with maximum hits of 50 and an E-value threshold of 10.

The site's Comparative Pathway metabolomics tool was used to gather data on the copy numbers of metabolic genes using KEGG pathway maps. The respective heat maps were compiled using GraphPad Prism version 8.4.3.

**Animals use and housing.** C57BL/6 mice were purchased from The Jackson Laboratory and used to establish an inbred murine colony in the conventional animal housing facility at Northwestern University. Pregnant dams were given a standard diet and water *ad libitum* and had full-term, vaginal births prior to experiments. Pups were allowed access to the dam's milk until weaning, when they were given standard mouse chow and water *ad libitum*. All animal procedures and experiments were performed under guidelines approved by the Institutional Animal Care and Use Committee at Northwestern University.

**Early life acquisition model.** An early-life acquisition model was developed to assess the colonization burden of *E. coli* strains. Two- to three-day-old littermates were randomly mixed between experimental cages and inoculated with 10 $\mu$L of $1 \times 10^6$ CFU bacteria ($n = 9$ to 12 pups, at least two cages per strain). For the inoculum, overnight cultures of bacterial cells were washed in PBS before being diluted to $1 \times 10^6$ CFU bacteria. All cells were diluted and plated on LB agar plates to ensure that pups received an equal inoculum bacterial CFU. Inoculum was given to pups orally using a pipet tip. Weights were taken twice weekly, and cages were changed daily after 10 days to avoid coprophagy. Stool was collected from individual mice at 3 and 4 weeks of age and homogenized in 500 mL of PBS for 1 min using a Fisherbrand Homogenizer probe, and bacterial populations were enumerated by plating on selective MacConkey agar. Bacterial CFU/g of stool was calculated.

To assess background Enterobacterales colonization among the mouse pups, stool samples from pups that did not undergo any manipulation were also plated on MacConkey agar with and without ceftriaxone.

**Competition experiments.** We chose ST410, ST28 (ESBL *E. coli* strains that colonized well), and ST1201 (ESBL *E. coli* that did not colonize well) and conducted *in vivo* competition experiments to look at their colonization potential compared to the commensal non-ESBL *E. coli* strain ST6438. The method followed the early-life acquisition model outlined above, but pups were inoculated with a 1:1 mixture of both competitor strains. Stool pellets were collected from each individual mouse at 3 and 4 weeks of life, and bacterial populations were enumerated following the early life acquisition model.

**Statistical analysis.** A Kruskal-Wallis test was used to compare bacterial CFU/g of stool between *E. coli* strains. *Post hoc* pairwise comparisons were made using Dunn's test. A Mann-Whitney-Wilcoxon test was used to compare bacterial CFU/g stool in the *in vivo* competition experiment. Statistical tests were performed in R version 4.0.0.

**Data availability.** All assembled genomes were submitted to National Center for Biotechnology Information (NCBI). The accession numbers of all of the strains can be found in Table S1.

## SUPPLEMENTAL MATERIAL

Supplemental material is available online only.
**SUPPLEMENTAL FILE 1**, PDF file, 0.1 MB.

## ACKNOWLEDGMENTS

This work was funded through the support provided to M.A. by NIH NIAID K08 AI123524 and the Duke Strong Start Award.

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
