## [Reviewer comments · Microbiology Spectrum]

Microbiology Spectrum

ESBL *Escherichia coli* isolates have enhanced gut colonization capacity compared to non-ESBL strains in neonatal mice

Aspen Kremer, Grant whitmer, Alondra Diaz, Alima Sajwani, Alexis Navarro, and Mehreen Arshad

Corresponding Author(s): Mehreen Arshad, Lurie Children's Hospital

Review Timeline:

Submission Date:	February 17, 2022
Editorial Decision:	April 13, 2022
Revision Received:	June 28, 2022
Editorial Decision:	July 12, 2022
Revision Received:	August 15, 2022
Accepted:	August 23, 2022

Editor: Sandeep Tamber

Reviewer(s): The reviewers have opted to remain anonymous.

Transaction Report:

DOI: <https://doi.org/10.1128/spectrum.00582-22>

April 13, 2022

Dr. Mehreen Arshad
Lurie Children's Hospital
Pediatrics
225 E. Chicago Ave
Chicago 60611

Re: Spectrum00582-22 (ESBL *Escherichia coli* isolates have enhanced gut colonization capacity compared to non-ESBL strains in neonatal mice)

Dear Dr. Mehreen Arshad:

Link Not Available

Sincerely,

Sandeep Tamber

Journals Department
Editor comments:

The methods need more elaboration and the discussion should explain the significance of the findings. In addition to the reviewer comments, please:

Describe how the virulence genes were identified. What search parameters were used for the BLAST analyses, and what were the criteria for a match?

Were the ESBL UPEC and non-ESBL strain isolated as a part of this study? Describe the method or state how they were obtained.

Lines 139-151: The identification of metabolic genes can go in the Results. The Discussion should explain the significance of the result.

Line 153: This conclusion is unsupported, please describe the diet and dietary changes in the methods. Explicitly explain the result in the Results and the significance in the Discussion.

Line 156: change "utilize" to "and have virulence factor genes"

Throughout: *E. coli* (italicized) as opposed to E. Coli

Reviewer comments:

Reviewer #1 (Comments for the Author):

Authors used whole genome sequencing to characterize fifteen ESBL *E. coli* strains isolated from infants in a Pakistani community. They also demonstrated significantly higher burden of intestinal colonization of some of the strains compared to two non-ESBL *E. coli* strains using a mouse model of early life challenge. Authors suggested that the ESBL strains may have enhanced fitness despite being highly antimicrobial resistant. To increase the significance of the study, it is recommended that authors select a few of the ESBL strains to conduct in vivo competition assays with one or both of the non-ESBL strains. Otherwise, authors may consider providing information on background gut microbiome of the dams to strengthen the study, as pups reared by different dams may have different gut microbiome that may influence their susceptibility to *E. coli* colonization. Authors may also consider addressing the specific comments below.

1. Methods, Sample Collection and Selective Markers, lines 72-74, please include information on both the ST6438 and the MG1655 non-ESBL strain, and specify what selective marker was used for culturing each strain.
2. Methods, Whole Genome Sequencing and Analysis, are the sequencing data deposited in public accessible database? If yes, please provide accession numbers.
3. Methods, Animal Model, please provide information on mouse strain, source and maintenance conditions. Please provide information on procedures for inoculation of 2 to 3 day-old pups and mixing of the pups. Please also provide information on stool collection, if it was done by collecting from individual mouse or as pool samples from the cages.
4. Methods, Animal Model, was there a control group of pups without inoculation of any *E. coli* strains? If not, it is highly recommended that authors include such a control group for investigation of the background *E. coli* and maybe Enterobacteriaceae in the mouse gut microbiota, in order to increase the certainty of recovering the inoculated *E. coli* stains.

Reviewer #2 (Comments for the Author):

This is a well-written manuscript on a most important topic. Below are some comments on the manuscript:

Abstract:

1. Row 28 : Extended spectrum beta lactamases is written differently from the introduction row 55.
2. Row 29: The pediatric age group is a broad. Narrow it down to neonates or infants or patients with an impaired developed immune-system would be more appropriate

Importance:

3. Row 44: Please, expand the discussion of the clinical relevance of ABR. Many patient groups except infants can be added.
4. Also expand the importance in risk infants such as preterms neonates, fullterm neonates in intensive care units and describe the difference to the healthy infant (up to 6 months).
The goal is to make the treating pediatricians and neonatologists to become aware of what child who is a risk patient for invasive disease with a virulent ESBL-producing strain.

Introduction:

5. Row 56: Enterobacteriaceae is now named as Enterobacterales. Can be corrected.

6. What is your explanation to the fact that certain strains resulted in increased burden of colonization?

Methods:

7. The study design is not mentioned
8. Where the strains collected in Pakistan and also analyzed in Paistan? Or transported to Chicago, USA? In that case,

information should be added. If transported, for how long stored. If so, that should be mentioned in the discussion part whether or not virulence and resistance genes can have been lost due to long storage, transportation and so on.

9. Row 70 Unclear from whom the samples were collected.

The patient group must shortly be described.

10. Does the authors know how many days of life the child was? Antenatal/perinatal/postnatal data on antibiotic use? It is of great interest to know whether or not the child has had an invasive disease caused by the ESBL producing EC strain.

11. Row 71: Name the method with which the sequence typing was performed by and the name of the laboratory.

12. Row 78. Remove the period after "(Promega, Madison, WI, USA)."

13. Row 84: Put the shortening MLST in parenthesis in the previous sentence

14. Row 98: Please describe if the pups were fullterm or premature born. Mode of birth? Also describe their nutrition during the study. Mouse milk? Other formula?

15. Row 101 and 103. "CFU/g of stool was calculated". CFU bacteria/g of stool?

Results:

16. In the method part it was mentioned that the rats were weighed repeatedly. Even though it was a small cohort, was there any significant difference in the mice with certain strains during the study period?

17. Did any of the mice develop invasive disease?

Discussion:

18. Discuss clinical relevance for the neonatologist/pediatrician dealing with non-ESBL/ESBL-producing Enterobacteriales.

19. Discuss the clinical relevance of knowing the metabolic genotype of the ESBL-producer

20. Discuss whether or not strains from healthy infants was right to choose and not strains that has caused disease in the infant?

Staff Comments:

Preparing Revision Guidelines

Please return the manuscript within 60 days; if you cannot complete the modification within this time period, please contact me. If you do not wish to modify the manuscript and prefer to submit it to another journal, please notify me of your decision immediately so that the manuscript may be formally withdrawn from consideration by Microbiology Spectrum.

Corresponding authors may join or renew ASM membership to obtain discounts on publication fees. Need to upgrade your

membership level? Please contact Customer Service at Service@asmusa.org.

Comments to the author:

This is a well-written manuscript on a most important topic. Below are some comments on the manuscript:

Abstract:

1. Row 28 : Extended spectrum beta lactamases is written differently from the introduction row 55.
2. Row 29: The pediatric age group is a broad. Narrow it down to neonates or infants or patients with an impaired developed immune-ssytem would be more appropriate

Importance:

3. Row 44: Please, expand the discussion of the clinical relevance of ABR. Many patient groups except infants can be added.
4. Also expand the importance in risk infants such as preterms neonates, fullterm neonates in intensive care units and describe the difference to the healthy infant (up to 6 months).
The goal is to make the treating pediatricians and neonatologists to become aware of what child who is a risk patient for invasive disease with a virulent ESBL-producing strain.

Introduction:

5. Row 56: Enterobacteriaceae is now named as Enterobacterales. Can be corrected.
6. What is your explanation to the fact that certain strains resulted in increased burden of colonization?

Methods:

7. The study design is not mentioned
8. Where the strains collected in Pakistan and also analyzed in Paistan? Or transported to Chicago, USA? In that case, information should be added. If transported, for how long stored. If so, that should be mentioned in the discussion part wheter or not virulence and resistance genes can have been lost due to long storage, transportation and so on.
9. Row 70 Unclear from whom the samples were collected.
The patient group must shortly be described.
10. Does the authors know how many days of life the child was?
Antenatal/perinatal/postnatal data on antibiotic use? It is of great interest to know wheter or not the child has had an invasive disease caused by the ESBL producing EC strain.
11. Row 71: Name the method with which the sequence typing was performed by and the name of the laboratory.

12. Row 78. Remove the period after "*Promega, Madison, WI, USA*."
13. Row 84: Put the shortening MLST in parenthesis in the previous sentence
14. Row 98: Please describe if the pups were fullterm or premature born. Mode of birth?
Also describe their nutrition during the study. Mouse milk? Other formula?
15. Row 101 and 103. "CFU/g of stool was calculated". CFU bacteria/g of stool?

Results:

16. In the method part it was mentioned that the rats were weighed repeatedly. Even though it was a small cohort, was there any significant difference in the mice with certain strains during the study period?
17. Did any of the mice develop invasive disease?

Discussion:

18. Discuss clinical relevance for the neonatologist/pediatrician dealing with non-ESBL/ESBL-producing Enterobacterales.
19. Discuss the clinical relevance of knowing the metabolic genotype of the ESBL-producer
20. Discuss whether or not strains from healthy infants was right to choose and not strains that has caused disease in the infant?

Reviewer Comment Response

Editor comments:

The methods need more elaboration, and the discussion should explain the significance of the findings. In addition to the reviewer comments, please:

Describe how the virulence genes were identified. What search parameters were used for the BLAST analyses, and what were the criteria for a match? Details of the analysis done to assess presence of known colonization factors has been added in the methods section under the heading 'Identification of resistance, virulence, and metabolic genes'.

Were the ESBL UPEC and non-ESBL strain isolated as a part of this study? Describe the method or state how they were obtained. We have expanded the description of the cross-sectional clinical study that we have previously conducted. We have explained how the strains were obtained, and that the commensal strain and ESBL strains were collected from the same infant cohort, though not the same infant.

Lines 139-151: The identification of metabolic genes can go in the Results. The Discussion should explain the significance of the result. We have added the figure in the main manuscript, and added more details in both the discussion and significance section.

Line 153: This conclusion is unsupported, please describe the diet and dietary changes in the methods. Explicitly explain the result in the Results and the significance in the Discussion. We have made explicit reference to the dietary change in the animals and added relevant references regarding the expected change in microbiome at the time of weaning.

Line 156: change "utilize" to "and have virulence factor genes". This change has been made.

Throughout: *E. coli* (italicized) as opposed to E. Coli. This correction has been done.

Reviewer comments:

Reviewer #1 (Comments for the Author):

Authors used whole genome sequencing to characterize fifteen ESBL *E. coli* strains isolated from infants in a Pakistani community. They also demonstrated significantly higher burden of intestinal colonization of some of the strains compared to two non-ESBL *E. coli* strains using a mouse model of early life challenge. Authors suggested that the ESBL strains may have enhanced fitness despite being highly antimicrobial resistant. To increase the significance of the study, it is recommended that authors select a few of the ESBL strains to conduct *in vivo* competition assays with one or both of the non-ESBL strains. Otherwise, authors may consider providing information on background gut microbiome of the dams to strengthen the study, as pups reared by different dams may have different gut microbiome that may influence their susceptibility to *E. coli* colonization. Authors may also consider addressing the specific comments below.

We thank the reviewer for these comments. We took the suggestion to conduct *in vivo* competition experiments and present that data in Figure 4.

1. Methods, Sample Collection and Selective Markers, lines 72-74, please include information on both the ST6438 and the MG1655 non-ESBL strain, and specify what selective marker was used for culturing each strain. Additional information on both strains has been added in the Methods section.

2. Methods, Whole Genome Sequencing and Analysis, are the sequencing data deposited in public accessible database? If yes, please provide accession numbers. We recently submitted all the whole genome sequencing data to NCBI (submission ID SUB11585770). But have not received accession numbers yet.

3. Methods, Animal Model, please provide information on mouse strain, source and maintenance conditions. Please provide information on procedures for inoculation of 2 to 3 day-old pups and mixing of the pups. Please also provide information on stool collection, if it was done by collecting from individual mouse or as pool samples from the cages. Details of the animals used, the housing/feeding conditions and the early life model has been added to the methods section.

4. Methods, Animal Model, was there a control group of pups without inoculation of any *E. coli* strains? If not, it is highly recommended that authors include such a control group for investigation of the background *E. coli* and maybe Enterobacteriaceae in the mouse gut microbiota, in order to increase the certainty of recovering the inoculated *E. coli* stains. This information has been added in text in both the methods and results section. We report that we were not able to isolate any bacterial colonies on MacConkey from pups which is not surprising given that *E. coli* is not a usual colonizer of mice.

Reviewer #2 (Comments for the Author):

This is a well-written manuscript on a most important topic. Below are some comments on the manuscript:

We thank the reviewer for these comments and appreciate the feedback.

Abstract:

1. Row 28 : Extended spectrum beta lactamases is written differently from the introduction row 55. This error has been corrected.

2. Row 29: The pediatric age group is a broad. Narrow it down to neonates or infants or patients with an impaired developed immune-system would be more appropriate. This change has been made.

Importance:

3. Row 44: Please, expand the discussion of the clinical relevance of ABR. Many patient groups except infants can be added. This has been added.

4. Also expand the importance in risk infants such as preterms neonates, fullterm neonates in intensive care units and describe the difference to the healthy infant (up to 6 months). This has been added.

The goal is to make the treating pediatricians and neonatologists to become aware of what child who is a risk patient for invasive disease with a virulent ESBL-producing strain.

Introduction:

5. Row 56: Enterobacteriaceae is now named as Enterobacterales. Can be corrected. We have made this correction.
6. What is your explanation to the fact that certain strains resulted in increased burden of colonization? A sentence has been added at the end of the introduction on our postulation.

Methods:

7. The study design is not mentioned The study design of the original study has been mentioned.
8. Where the strains collected in Pakistan and also analyzed in Pakistan? Or transported to Chicago, USA? In that case, information should be added. If transported, for how long stored. If so, that should be mentioned in the discussion part whether or not virulence and resistance genes can have been lost due to long storage, transportation and so on. These strains were collected in Pakistan and then transported to the US. We have added information on the experiments which we conducted to assess the stability of the ESBL gene. And also discuss this in the discussion section. It is important to note that the strains were sequence at roughly the same time as the animal experiments were conducted.
9. Row 70 Unclear from whom the samples were collected. The patient group must shortly be described. This has been clarified in the methods section.
10. Does the authors know how many days of life the child was? Antenatal/perinatal/postnatal data on antibiotic use? It is of great interest to know whether or not the child has had an invasive disease caused by the ESBL producing EC strain. More details regarding the clinical study have been added.
11. Row 71: Name the method with which the sequence typing was performed by and the name of the laboratory. This has been added.
12. Row 78. Remove the period after "(Promega, Madison, WI, USA)." This error has been corrected.
13. Row 84: Put the shortening MLST in parenthesis in the previous sentence. This change has been made as suggested.
14. Row 98: Please describe if the pups were fullterm or premature born. Mode of birth? Also describe their nutrition during the study. Mouse milk? Other formula? The pups were full-term and were housed with the dams so had access to maternal milk ad lib. This has been added to the methods section.
15. Row 101 and 103. "CFU/g of stool was calculated". CFU bacteria/g of stool? This has been corrected.

Results:

16. In the method part it was mentioned that the rats were weighed repeatedly. Even though it was a small cohort, was there any significant difference in the mice with certain strains during the study period? There was no difference and that has been noted in the text in the results section.
17. Did any of the mice develop invasive disease? No, that has been noted in the results section as well.

Discussion:

18. Discuss clinical relevance for the neonatologist/pediatrician dealing with non-ESBL/ESBL-producing Enterobacterales. This has been added to the first paragraph of the discussion.

19. Discuss the clinical relevance of knowing the metabolic genotype of the ESBL-producer. This has been added to the para 5 of the discussion.

20. Discuss whether or not strains from healthy infants was right to choose and not strains that has caused disease in the infant? This has been added to the first paragraph of the discussion.

July 12, 2022

Dr. Mehreen Arshad
Lurie Children's Hospital
Pediatrics
225 E. Chicago Ave
Chicago 60611

Re: Spectrum00582-22R1 (ESBL *Escherichia coli* isolates have enhanced gut colonization capacity compared to non-ESBL strains in neonatal mice)

Dear Dr. Mehreen Arshad:

Thank you for submitting your manuscript to Microbiology Spectrum. As you will see your paper is very close to acceptance. Please modify the manuscript along the lines I have recommended. As these revisions are quite minor, I expect that you should be able to turn in the revised paper in less than 30 days, if not sooner. If your manuscript was reviewed, you will find the reviewers' comments below.

When submitting the revised version of your paper, please provide (1) point-by-point responses to the issues I raised in your cover letter, and (2) a PDF file that indicates the changes from the original submission (by highlighting or underlining the changes) as file type "Marked Up Manuscript - For Review Only". Please use this link to submit your revised manuscript. Detailed instructions on submitting your revised paper are below.

Link Not Available

Sincerely,

Sandeep Tamber

Editor comments:

- there needs to be a data availability statement in the methods indicating the accession numbers of the sequenced strains
- L79: please state the selective agar used
- L112 - 114: Please ensure that the relevant references are cited for the use of Enterobase, PATRIC, and CARD
- It is not clear how many isolates were collected and STs were found. It appears as 17 isolates and 14 STs in Figure 1. L164 states 12 STs were found. L228 states that 11 STs were found in 15 isolates. Please state clearly how many strains were collected and how many STs were found.

Reviewer comments:

Reviewer #1 (Comments for the Author):

There may be an error in line 97, "in vivo" may need to be changed to "in vitro".

Preparing Revision Guidelines

- point-by-point responses to the issues I raised in your cover letter
- Upload a compare copy of the manuscript (without figures) as a "Marked-Up Manuscript" file.
- Each figure must be uploaded as a separate file, and any multipanel figures must be assembled into one file.
- Manuscript: A .DOC version of the revised manuscript
- Figures: Editable, high-resolution, individual figure files are required at revision, TIFF or EPS files are preferred

Please return the manuscript within 60 days; if you cannot complete the modification within this time period, please contact me. If you do not wish to modify the manuscript and prefer to submit it to another journal, please notify me of your decision immediately so that the manuscript may be formally withdrawn from consideration by Microbiology Spectrum.

Reviewer Comment Response

Editor comments:

- there needs to be a data availability statement in the methods indicating the accession numbers of the sequenced strains. This has been added to both the methods section under 'Whole Genome Sequencing and Analysis' and in the first paragraph of the results section.
- L79: please state the selective agar used This has been added.
- L112 - 114: Please ensure that the relevant references are cited for the use of Enterobase, PATRIC, and CARD These have been added.
- It is not clear how many isolates were collected and STs were found. It appears as 17 isolates and 14 STs in Figure 1. L164 states 12 STs were found. L228 states that 11 STs were found in 15 isolates. Please state clearly how many strains were collected and how many STs were found. We have corrected the number in both paragraphs.

Reviewer comments:

There may be an error in line 97, "in vivo" may need to be changed to "in vitro". This has been corrected.

August 23, 2022

Dr. Mehreen Arshad
Lurie Children's Hospital
Pediatrics
225 E. Chicago Ave
Chicago 60611

Re: Spectrum00582-22R2 (ESBL *Escherichia coli* isolates have enhanced gut colonization capacity compared to non-ESBL strains in neonatal mice)

Dear Dr. Mehreen Arshad:

Your manuscript has been accepted, and I am forwarding it to the ASM Journals Department for publication. You will be notified when your proofs are ready to be viewed.

Sincerely,

Sandeep Tamber
Editor, Microbiology Spectrum
